# Pregnancy Apps for Self-Monitoring: Scoping Review of the Most Popular Global Apps Available in Australia

**DOI:** 10.3390/ijerph20021012

**Published:** 2023-01-05

**Authors:** Natasa Lazarevic, Marie Lecoq, Céline Bœhm, Corinne Caillaud

**Affiliations:** 1Biomedical Informatics and Digital Health, School of Medical Sciences, Faculty of Medicine and Health, The University of Sydney, Camperdown, NSW 2006, Australia; 2Charles Perkins Centre, The University of Sydney, Camperdown, NSW 2006, Australia; 3AgroParisTech, Universite Paris-Saclay, CEDEX, 91 123 Palaiseau, France; 4School of Physics, Faculty of Science, Physics Building, The University of Sydney, Camperdown, NSW 2006, Australia

**Keywords:** digital health, pregnancy, smartphone apps, mobile phone, data security, scorecard

## Abstract

Digital health tools, such as apps, have the potential to promote healthy behaviours, especially self-monitoring, which can facilitate pregnancy management and reduce the risk of associated pregnancy health conditions. While pregnancy apps are popular amongst pregnant women, there is limited information about the overall quality of their content or self-monitoring tools and the number of behaviour change techniques (BCTs) that they include. The aim of this study was thus to assess the quality of pregnancy apps for self-monitoring, and their usage of BCTs. We identified pregnancy apps by web scraping the most popular global apps for self-monitoring in the Apple App Store and Google Play Store available in Australia. The app quality was evaluated using the scorecard approach and the inclusion of BCTs was evaluated using the ABACUS tool. We identified 31 pregnancy apps that met our eligibility criteria. We found that pregnancy apps tended to score the highest in the domains of ‘cost and time’, ‘usability’, and ‘technical’, and lowest on ‘clinical’ and ‘end-user requirements’. Additionally, the majority of apps contained minimal BCTs. Based on our findings, we propose a digital health scorecard visualisation that would break down app quality criteria and present them in a more accessible way to clinicians and pregnant users. We conclude that these findings highlight the shortcomings of available commercial pregnancy apps and the utility of a digital health scorecard visualisation that would empower users to make more informed decisions about which apps are the most appropriate for their needs.

## 1. Introduction

It is estimated that there are 5.48 billion smartphone users with access to high-speed Internet (68% of the global population) around the world [1]. This increased market penetration of smartphones presents an opportunity to improve access to, and the quality of, healthcare. One sector that would significantly benefit from digital healthcare is pregnancy as it could provide expectant mothers with reassurance and, with close monitoring, could signal the need for emergency care [2]. This is especially relevant for people living in remote, rural, or disadvantaged communities where access to healthcare is limited and the risk of adverse pregnancy outcomes such as perinatal death is higher [3,4,5,6,7,8]. Exacerbated by the COVID-19 pandemic, mobile apps are being increasingly used by pregnant women to access pregnancy-related information as well as for self-monitoring [9,10,11,12,13,14,15]. In fact, 6 pregnancy apps are listed in the top 100 most popular medical apps available on the Apple App Store [16].

However, pregnancy apps are not being utilised to their full potential. Studies have shown that pregnancy apps could be improved to include: (i) more credible information, (ii) features for self-monitoring outside of just including general information about foetal development, and (iii) features to connect users to healthcare professionals, facilitating access to healthcare [17,18,19]. Ultimately, if used in conjunction with appropriate remote or in-person healthcare, pregnancy apps could help to prevent adverse outcomes associated with pregnancy. In addition, mobile apps offer the opportunity for patients to self-monitor health measures such as blood pressure, weight, and blood glucose, which would allow for pregnancy risk assessment and, with appropriate oversight from healthcare professionals, can signal a need for intervention before or after the development of health conditions during pregnancy [20]. Indeed, digital self-monitoring tools have been shown to be feasible for blood pressure monitoring [21], to encourage healthy eating behaviours [22], and to improve pregnancy outcomes such as physical activity and weight gain during pregnancy [23].

Several recent pregnancy app scoping reviews assessing the content quality of general health information [18,24,25], guidance for decreased foetal movements [26], improving pregnancy outcomes [19,27], self-monitoring [28], gestational weight gain tracking [29], physical activity [30,31], nutrition/diet information [32,33,34,35], and anxiety and mental health [36,37] all overwhelming emphasised the need for better content quality, even though most apps had a good user experience design and aesthetic features. Several of these studies also highlighted the lack of appropriate app behaviour change techniques, and the need for better self-monitoring or health tracking [24,29,33,34,35].

However, none of these studies are yet to systematically evaluate the prevalence and quality of self-monitoring tools and behaviour change techniques (BCTs) contained within pregnancy apps that are focused on monitoring physical health and wellness. Several systematic reviews have demonstrated that the inclusion of BCTs increases adherence, engagement, and the effectiveness of pregnancy digital health interventions [38,39]. Further, the inclusion of techniques to change health-related behaviours in apps can play a major role in reducing a person’s risk of developing a health condition and can help a person to manage their condition [40]. The inclusion of BCTs in apps has also been shown to promote healthy behaviours during pregnancy. For instance, the Healthy Motivations for Moms-To-Be intervention study found that using behaviour change techniques of goal-setting and self-monitoring in an app promoted healthy behaviours, especially for long-term healthy eating [22]. Thus, apps should not only include credible information but BCTs to promote self-monitoring behaviours. 

Further, the majority of the published pregnancy scoping reviews use the Mobile App Rating Scale (MARS) and/or newly created scales to assess app quality [28,29,32,34]. However, these assessment tools do not incorporate the needs or requirements of end-users and are limited in their technical, user experience, and interface assessment. In response, Matthews et al. (2019) [41] recently proposed a “validation framework for mobile apps” that takes into account (i) recent advances in technology, (ii) the clinical risks associated with the use of poor-quality health apps, and (iii) the need for an independent evaluation of digital health products before they are released into the market. The practical application of this framework, the “Digital Health Scorecard” approach, is the app quality assessment tool proposed to evaluate commercial apps. The scoring criteria used in this approach were designed by experts in their respective field (e.g., app developers defining the usability criteria) [42]. 

The majority of the published pregnancy app reviews identify and collect app data manually [19,24,25,26,27,28,30,31,34,35,37]. A recent review of mobile apps for women with anxiety in pregnancy used a novel approach for the identification of apps based on defined search terms by using an open-source web-scraping method [36]. 

In this study, we used a novel approach by combining web scraping to identify relevant apps with the scorecard approach to assess the app quality. The goal of this study was thus to assess (i) the quality of pregnancy self-monitoring apps and (ii) the usage of behaviour-change techniques by apps to promote self-monitoring, and (iii) to compare these evaluations to global app ratings. We found that popular pregnancy apps for self-monitoring had low overall app quality scores, especially for their clinical content and behaviour change techniques, and did not meet the requirements of end-users. Additionally, there were significant positive correlations between: (1) app quality scores and the number of monitoring tools and (2) our scorecard ratings and the global app ratings. These findings highlight the shortcomings of available commercial pregnancy apps, and we propose a “Digital Health Scorecard” visualisation to help end-users identify app quality and improve their decisions about which digital health tools to use. 

## 2. Materials and Methods

### 2.1. Study Design

This study is a scoping review of the most popular pregnancy apps on the global market that were available in Australia from May 2021 to June 2021. The pregnancy apps were assessed for their quality, content, self-monitoring tools, and behaviour change techniques. The review was conducted by following the Preferred Reporting Items for Systematic Reviews—Scoping Reviews (PRISMA-Scr) guidelines [43]. Additionally, the Quality and Risk of Bias checklist for Studies That Review Smartphone Applications was used when reporting the methods [44]. 

### 2.2. Step 1: Identification of Smartphone Apps

Google Play Store and Apple App Store were web scraped in May-June 2021 from the global stores in the United States (refer to Appendix A for the full code used for the web scraping). The global stores were chosen to be scraped as they represent the largest market for pregnancy apps, and the global apps available in Australia were included in this review. An open-source web-scraping method written in JavaScript as a Node.js module was used since comprehensive data from app stores could be automatically retrieved. Using this web-scraping method, the search parameters to retrieve apps can be defined (such as the country’s app store and the language the app is in), and multiple search terms can be scraped at once [45,46]. For each search term, the top 250 apps in Google Play Store and the top 200 apps in Apple App Store were retrieved. 

Apps were identified by using the following search terms: ‘pregnancy’, ‘pregnant’, ‘pregnant woman’, ‘pregnant women’, ‘gestation’, ‘prenatal’, ‘baby’, ‘babies’, ‘foetus’, ‘fetus’, ‘gynecology’, ‘gynaecology’, ‘obstetrics’, ‘OBGYN’, ‘pregnancy monitor’, ‘pregnancy tracker’, ‘contractions’, ‘pregnancy weight’, ‘pregnancy BMI’, ‘pregnancy education’, ‘pregnancy support’, ‘pregnancy community’, ‘gestational diabetes’, ‘preeclampsia’, ‘pregnancy ultrasound’, ‘pregnancy images’, ‘pregnancy photos’. Additionally, apps were also identified by searching the store categories so as not to omit apps that were not extracted by searching the main store databases. The 200 most popular app search results were scraped from categories on both stores, which included the medical, education, lifestyle, health, and fitness categories (the parenting category only exists in Google Play Store). 

Scraping was run on Visual Studio Code (Version 1.72.2). Each app’s app ID, title, URL, description, icon, genre, content rating, available languages, storage size, required version of device, date of release, date of last update, release notes, app version, price, developer ID, name of developer, developer URL, app rating, app reviews, number of installs, privacy policy, and which devices can support the app were retrieved if available. The app rating and developer data were updated in October 2022. 

### 2.3. Step 2: Screening of Smartphone Apps

App results scraped for each search term were assessed for eligibility for inclusion in this study. Python scripts and excel commands were used to remove (i) duplicates, (ii) apps with a user rating below 4, and (iii) apps that had fewer than 20 user ratings. Eligibility was then further assessed manually, and apps were only included if they (i) were in English, (ii) were designed for pregnant women, (iii) included more than one monitoring tool (most kick or contraction counter apps were excluded), and (iv) included a component on monitoring physical health or wellness during pregnancy. Free, freemium, and paid apps were included in this review. Premium accounts were accessed using a free trial where possible (Table 1). Subscriptions for premium accounts were paid for if they enabled access to an extra health monitoring feature. Whether apps required or offered subscriptions was recorded at the time of download. The number and types of monitoring tools apps offered was recorded. Apple Store apps were all reviewed on iOS mobile devices and Google Play Store were reviewed on an Android mobile and an Android tablet device.

### 2.4. Step 3: Evaluation of Smartphone Apps

The digital health “scorecard approach” was used to score and analyse the apps for their content quality [41,42]. This approach was selected as it incorporates conventional app quality assessment criteria (content quality) with end-user requirements and technical features. The “scorecard approach” assessment criteria are divided into 5 domains: (1) technical, (2) clinical, (3) usability, (4) cost and time, and (5) end-user requirements. The same criteria outlined by [42] were used, with minor modifications to domains 1, 2, 4, and 5. Refer to Appendix A for detailed modified criteria. With respect to cost and time domain, apps were given a ‘cost’ score based on their price for download or subscription and whether they included advertisements and a ‘time’ score based on how long it took to complete onboarding, how long it took to learn to navigate the app, and whether additional training was required. The end-user requirements domain was modified to reflect the needs of pregnant women and healthcare professionals. These requirements were developed by referring to published articles that investigated user needs of pregnant women [47] and midwives [48]. Several assessment criteria for end-user requirements outlined by Sedhom et al. (2021) [42] were retained as they reflected the needs of end-users (Appendix A). In addition to the 5 domains outlined above, app behaviour change techniques (BCTs) were evaluated using the App Behaviour Change Scale (ABACUS) [49]. ABACUS is divided into 4 assessment criteria: (1) knowledge and information, (2) goals and planning, (3) feedback and monitoring, and (4) actions (Appendix A). 

The apps were downloaded and apps available during the entire screening period were scored (domain scores and BCT scores) independently by at least two reviewers (N.L., M.L., and C.C.). The reviewing team included an exercise physiologist and digital health expert (C.C.), a PhD student with a digital health, technical, user experience, and interface background (N.L.), and a student with a computer engineering background (M.L.). If any uncertainty arose during scoring, reviewers discussed it with another researcher (C.B., who has a technical and software engineer background). Apps were left running in the background during scoring to allow for assessment of reminders and notifications. Inter-scorer reliability was evaluated by running an intraclass correlation across multiple scorecard and BCT domains and was found to be 0.72 (F_359,355_ = 6.2, *p* < 0.001). An intraclass correlation of less than 0.5 indicates poor reliability, between 0.5 and 0.75 indicates moderate reliability, between 0.75–0.9 indicates good reliability, and above 0.9 indicates excellent reliability. The scoring criteria were also discussed during several stages of scoring to ensure all scorers interpreted the criteria consistently. 

### 2.5. Step 4: Scoring and Analysis of Smartphone Apps

For each app, a scorecard score was calculated as an average of the app’s total domain score and BCT score. As outlined by Sedhom et al. (2021) [42], to calculate the total domain score, the app’s individual domain scores were averaged (equally weighted). To calculate an individual domain score (i.e., (1) technical, (2) clinical, (3) usability, (4) cost and time, and (5) end-user requirements), the subdomain scores were averaged (equally weighted) across all subdomains. Each subdomain score was calculated as a percentage of the maximum possible score. An analogous procedure was used to calculate each app’s BCT score. This method is described by the following equations: (1)$$\mathrm{Individual}\;\mathrm{Domain}\;\mathrm{Score}=\frac{\left(\mathrm{Subdomain}\;1\;\left(\%\right)+\;\mathrm{Subdomain}\;2\;\left(\%\right)+\ldots +\;\mathrm{Subdomain}\;n\;\left(\%\right)\right)}{n}$$
(2)$$\mathrm{Total}\;\mathrm{Domain}\;\mathrm{Score}=\frac{\left(\mathrm{Technical}\;+\;\mathrm{Clinical}\;+\;\mathrm{Usability}\;+\;\mathrm{Cost}\;\mathrm{and}\;\mathrm{time}\;+\;\mathrm{User}\;\mathrm{requirements}\right)}{5}$$
(3)$$\mathrm{BCT}\;\mathrm{Score}=\frac{\left(\mathrm{Knowledge}\;\mathrm{and}\;\mathrm{information}\;+\;\mathrm{Goals}\;\mathrm{and}\;\mathrm{planning}\;+\;\mathrm{Feedback}\;\mathrm{and}\;\mathrm{monitoring}\;+\;\mathrm{Actions}\right)}{4}$$
(4)$$\mathrm{Scorecard}\;\mathrm{Score}=\frac{\left(\mathrm{Domain}\;\mathrm{score}\;+\;\mathrm{BCT}\;\mathrm{score}\right)}{2}$$

### 2.6. Statistical Analysis

All statistical analysis was performed in GraphPad Prism v9.3.1. Pearson’s correlation coefficients were computed to assess the linear relationships between each of the individual domain scores with each other and with the global app ratings. Simple linear regressions were performed to assess the relationships between (i) total domain score and number of monitoring tools, and (ii) scorecard score and global app ratings. All graphs were created using GraphPad Prism and exported to Adobe Illustrator to finalise the figurework. 

## 3. Results

Our search strategy retrieved 12,483 results (main store database n = 10,683 and categories n = 1800). After removing duplicates, apps with a user rating < 4, and apps outside the scope, the remaining commercial apps (n = 224) were then screened based on their description and downloaded and assessed for eligibility (n = 91). A total of 4 of these apps were found to no longer have a user rating of 4 or above and were excluded. Finally, 31 apps were deemed eligible for inclusion in this review and were assessed for content analysis and quality and behaviour change techniques (Figure 1). Of those 31 apps, 12 were available on both stores, 9 were available only on Apple App Store, and 10 were available only on Google Play Store. The apps were listed in either the medical category (14 in Apple App Store and 7 in Google Play Store), in the parenting category (11 only in Google Play Store), or in the health and fitness category (7 in Apple App Store and 4 in Google Play Store). See Appendix A for a full list of the apps included for quality assessment. 

### 3.1. Self-Monitoring Tools

Across the 31 apps reviewed, 15 different monitoring tools were identified. The average number of monitoring tools included in each app was 5 (range 2–14). The breakdown of the monitoring tools contained within each app is shown in Table 2. Ovia Pregnancy Tracker (14/15) and GLOW/Nurture Pregnancy Tracker (10/15) included the most tools. The top 4 monitoring tools offered by pregnancy apps were weight tracker (27/31), contraction/kick counter (23/31), diary/journal (18/31), and bump tracking with a photo or measuring tape (16/31). Several apps included features within the monitoring sections of the app that were not relevant. This included six pregnancy apps with baby name finders and another six that offered an internal pregnancy/baby shop. The Pregnancy Tracker-Enes app included features for receiving messages from the user’s unborn baby and baby horoscopes. Apps that included 3D or virtual reality representations of the developing foetus did not allow skin colour to be customised. Aside from offering information about foetal development, the majority of the apps (20/31) shared no specific content and some diagrams about changes occurring to the pregnant woman’s body (such as the size of their uterus and changes to their hips/pelvis). Only five of the apps included a feature for reporting miscarriages, two of which were not explicit (e.g., “no longer pregnant” and “remove child from profile” buttons). 

### 3.2. Scorecard Approach

The average total domain score for all of the apps was 40.0%. The averages of each of the individual domain scores across all apps, in order of highest to lowest, were: cost and time (60.0%), usability (55.2%), technical (43.2%), end-user requirements (31.3%), and clinical (10.1%). There was a large variability in the individual domain scores across the 31 apps, except for the clinical domain (Figure 2a). A correlation matrix between each individual domain score and global app rating was calculated (Figure 2b). See Appendix A for full statistical results, including statistical significance, and the 95% confidence intervals. 

#### 3.2.1. Clinical

None of the apps made claims to impact clinical health outcomes. All apps asserted that they only provide general health tips and tracking. The clinical domain had the lowest individual domain score (10%), with 16/31 of the apps scoring 0%. The two apps that scored the highest were affiliated with an accredited medical institution or hospital. These were Woman’s Pregnancy app (63%), which is affiliated with the Woman’s hospital, and the WebMD pregnancy app (63%), which is affiliated with the WebMD medical institution. None of the health tips provided in any of the apps were supported by primary research articles, such as randomised, controlled trials. However, 16/31 of the apps shared health tips supported by “expert opinion”. A total of 12 apps were flagged during review for health tips that were false or misleading. Examples of misleading content included: the person listed as providing expert opinion was not credible; misleading guidelines and information related to taking body measures, such as weight; health information was phrased to be stress-inducing; health information was listed as “facts” without any evidence supporting the claims. 

#### 3.2.2. Technical

Apps that scored the highest in the technical domain included Pregnancy+ (83%), Glow/Nurture Tracker (79%), and Ovia Pregnancy Tracker (77%). Apps that scored the lowest in the technical domain included Happy Pregnancy Ticker (5%) and Babynote (13%). App technical scores on subdomains varied: security (29%), privacy (53%), interoperability (31%), and performance (61%). The majority of apps (20/31) share data to third parties that are linked to the identity of users (these could include data from contact info, contacts, usage data, user content, location, identifiers, financial info, sensitive info, search history, diagnostics, purchases, health and fitness, and other data). In total, 5/31 apps provided no details about privacy practices and the handling of data. Forty-five percent (14/31) of apps provided a means to export user data (4/31 exported data as PDFs and 10/31 exported data as CVSs and/or JSON). None of the apps provided information within the app or app store that they support Fast Healthcare Interoperability Resources (FHIR) application programming interfaces (APIs), which allow for user data to be integrated with other software, such as electronic medical records. 

#### 3.2.3. Usability—User Experience and Interface

Ovia Pregnancy Tracker (96%), Baby2Body (84%), Pregnancy+ (78%), and Velmio Pregnancy Tracker (77%) apps scored the highest in overall usability. Apps generally scored the highest on the ‘visual design’ (75%) and ‘app navigation’ (67%) subdomains. For the ‘context personalisation’ subdomain, the personalisation of content and screens was primarily updated only based on the users nominated pregnancy week. Only two apps (Ovia Pregnancy Tracker and Glow/Nurture Tracker) included personalisation based on the health data entered by users over time. Ovia Pregnancy Tracker also based personalisation on user responses to a health questionnaire. All 12 apps that were available on both stores had flexible interfaces (i.e., operable on both iOS and Android devices). 

#### 3.2.4. Cost and Time

All 31 apps were free to download. However, 4 of the apps (listed as free) required a payment or subscription to access the majority of the app features. Most apps (23/31) included advertisements when using the free version and the usability one app, Happy Pregnancy Tracker, was limited due to the higher number of advertisements. The average score for the ‘cost’ subdomain was 70%. The time taken to set-up the apps or onboarding took less than 2 min (8/31), 3–5 min (17/31), or more than 5 min (6/31). Some apps (15/31) required additional training to understand some of the app functionalities. Apps were also assessed on whether the user experience (UX) of the subscription screens were misleading (did it feel like you needed to subscribe to use the app even though it is free?), and 7/31 were found to have misleading UX for subscription plans either during onboarding or when reopening the app. The average score for the ‘time’ subdomain was 49%. 

#### 3.2.5. End-User Requirements

Ovia Pregnancy Tracker (69%), GLOW/Nurture Tracker (67%), and WebMD Pregnancy (57%) apps scored the highest for the end-user requirements domain. Across all apps, the average scores for the ‘education’, ‘tracking’, and ‘social support’ subdomains were 52%, 35%, and 30%, respectively. Apps scored especially poorly in the ‘prevention’ subdomain (9%), with apps providing minimal or no personalized risk assessment and risk mitigation techniques. One of the major end-user requirements identified for the ‘tracking’ subdomain was that the “app must monitor changes in my body”, and this form of monitoring was mainly performed by measuring weight and taking bump photos. However, apps rarely provided information regarding what these measurements meant, did not relate them to guidelines, and did not explain why users should monitor these measurements. Another end-user requirement in the ‘tracking’ subdomain was whether apps had a feature for users to connect or message their healthcare team. Only 26% of apps had a feature to connect users to their healthcare professionals primarily by allowing users to share a ‘health report’ generated by the app via email. Only one app, Ovia Pregnancy Tracker, had a feature for users to connect with their healthcare professional/provider if the user has the relevant health insurance plan and lives in the United States. 

A linear regression between the total domain score and the number of monitoring tools that the app contained found a significant correlation equation: [number of monitoring tools] = −0.89 + 0.15 × [Total Domain Score], R^2^ = 0.37, F_1,29_ = 16.77, *p* < 0.001, indicating that the greater number of monitoring tools that an app contained, the higher total domain score that the app received (Figure 3). 

### 3.3. Behaviour Change Techniques (BCTs)

The mean total BCT score for all apps was 21.1%. The averages of the individual categories, in order of highest to lowest, were: knowledge and information (37.9%), feedback and monitoring (21.7%), actions (13.6%), and goals and planning (11.3%). There was a large variability in the BCT category scores across the 31 apps, except for the ‘goals and planning’ (Figure 4a). Velmio Pregnancy Tracker (85%), Baby2Body (80%), and Pregnancy Tracker ~Fittur (75%) had the highest overall BCT score. In the ‘feedback and monitoring’ category, apps scored the highest in allowing users to easily self-monitor behaviour (71%). Apps, however, did not have features to export data (35%), share behaviours with others (15%), provide user feedback (11%), provide general encouragement (10%), allow users to understand differences quickly and easily between current actions and future goals (7%), and provide a material or social reward or incentive (3%) (Figure 4b).

### 3.4. Overall Quality of Pregnancy Apps

The total domain scores, BCT scores, and global app ratings for the top 10 pregnancy apps identified by their overall scorecard score are detailed in Table 3. A simple linear regression was calculated to determine if our overall scorecard score correlated with the app’s global app rating. A significant correlation equation was found: [global app rating] = 4.21 + 0.013 × [Scorecard Score], R^2^ = 0.28, F_1,29_ = 10.97, *p* = 0.003 (Figure 5). Refer to Appendix A for the total for the 31 apps included in this review.

## 4. Discussion

### 4.1. Summary of Findings

This scoping review aimed to: (i) assess the quality of the most popular commercial pregnancy self-monitoring apps using the novel scorecard approach developed by Mathews et al. (2019) [41]; (ii) assess the usage of behaviour change techniques by these apps to promote self-monitoring; and (iii) compare our evaluations to global user app ratings. We found that pregnancy self-monitoring apps generally had low overall scorecard scores (range 13.9–62.7%). Apps tended to score the highest in the domains of ‘cost and time’, ‘usability’, and ‘technical’, and lowest on ‘clinical’ and ‘end-user requirements’. Additionally, the majority of apps contained minimal behaviour change techniques (BCTs). Interestingly, the number of monitoring tools an app contained was positively correlated with the total domain score. Additionally, the total scorecard score (i.e., domain scores + BCT scores) was positively correlated with the app’s global user ratings on the Google Play Store and/or Apple App Store, indicating that users value many of the features assessed by our scorecard approach. These findings thus illustrate the utility of the “Digital Health Scorecard” approach and its potential to help end-users choose the most appropriate app for their needs by providing detailed information about app quality and functionality in an accessible way. 

### 4.2. What Factors Contribute to the Quality of Pregnancy Apps for Self-Monitoring?

The scorecard score results demonstrate that there is variable quality amongst the available commercial pregnancy apps for self-monitoring. The top 10 pregnancy apps that we identified using our scorecard approach had an average rating of 2.3/5, indicating that there is scope for improvement across several domains. Additionally, increasing the use of BCTs would significantly improve the overall app quality as defined by our total scorecard score. Specifically, apps included limited numbers of BCTs related to “feedback and monitoring”, “actions”, and “goals and planning” categories. These findings are consistent with a trend observed in prior pregnancy app reviews that apps tend to score higher for usability and lower for content quality and the inclusion of behaviour change techniques [24,25,28,29,30,31,32,34]. The domains and subdomains that we identified as having the largest scope for improvement are: ‘clinical’, ‘end-user requirements’, and ‘data privacy and security’. 

We found there to be a significant positive correlation between the number of monitoring tools and total domain scores, suggesting that the monitoring of more health parameters was associated with the overall app quality. However, in line with the findings of Musgrave et al. (2020) [19], we found that most apps did not explain why users should monitor the health parameters (such as the weight or blood pressure), how the parameters should be monitored, or how to interpret the results. Further, they did not relate the health results to credible guidelines, such as those outlined by governments and medical institutions. 

Additionally, even though apps often scored highly in the “technical” domain, they included poor privacy and security policies, especially regarding how much user data apps collect and share to third parties. Sensitive health information, such as if a woman is pregnant, whether they develop any associated conditions or complications, as well as the overall progress of the pregnancy, must be stored securely. This issue has been brought to the fore by the recent overturning of Roe v. Wade, which has catalysed a broader discourse around the need for better data security and privacy and quality control of women’s health apps to ensure that data are not shared with third parties, such as insurance companies. Alarmingly, we found that 65% of the pregnancy apps that we reviewed shared user data to third parties and that apps scored only 29% in the data security subdomain. Similarly, a recent review assessing the privacy, data sharing, and data security policies of 23 women’s mobile health apps found that 87% of apps shared user data to third parties and 87% of apps would share user data if required by law [50]. App users should thus be better informed about the associated privacy risks of using female health apps, and the App Store approval of apps should be more rigorous in relation to data security and privacy. The lack of control over data and transparency has had widespread impacts on people’s trust in health apps and emphasises the need for more human-centred design [51,52] and ownership of data [53,54]. 

The scorecard approach, unlike other app evaluation methods, incorporates end-user requirements in its assessment [55]. Our findings suggest that the vast majority of commercially available apps do not consider the experiences or needs of their pregnant users. For instance, such a low proportion of apps included features for reporting miscarriages, which is unsettling given how common and difficult miscarriages are [56]. As noted by Keep (2021) [57], miscarriages need to be talked about so that the appropriate evidence-based continuous care is provided. With privacy ensured, and codesign with users, apps could provide additional support to users who have experienced a miscarriage alongside the appropriate in-person care. These findings also support the findings of Frid et al. (2021) [24] that apps did not include content nor features that provided holistic support through all stages of pregnancy. Moreover, consistent with Lupton & Thomas (2015) [58], the majority of pregnancy apps that we reviewed reinforced stereotypes (especially those related to gender norms) and did not consider diverse pregnancy experiences, such as feeling ambivalent about the upcoming pregnancy. 

Thus, developers, researchers, and innovators should consider several factors to design higher quality apps for self-monitoring: (1) include higher numbers of self-monitoring tools and BCTS; (2) include more comprehensive data privacy and security policies, especially for data sharing; (3) higher content quality and usability; and (4) consider the experiences of pregnant users and their needs. 

### 4.3. How Can We Relay These Findings to Users to Inform Their App Choices?

There was a significant correlation between our total scorecard ratings and the app store user ratings. However, as noted by Bondaronek et al. (2018) [59], it is not entirely clear how app store ratings are calculated, but data related to user ratings (which could include fake ratings), reviews, downloads, and app usage data are likely aggregated. However, these app ratings have a great influence over which apps users choose to download [60]. If App Store user ratings do not represent an app’s true overall quality and utility, users are unable to make informed decisions. As such, there is a need for a credible scoring platform that is more accessible to end-users. Such a platform would likely perform a dual function. Firstly, it would allow users to choose which apps best suit their needs and preferences. Secondly, as outlined by Mathews et al. (2019) [41], a formal independent evaluation of digital health products would help to improve their overall quality by encouraging developers to address the range of domains before and after their release to the market. 

We suggest using a visualisation that includes a breakdown of the subdomain scores, BCT scores, and data from the app stores (e.g., app store rating, number of downloads, data security, and privacy information). Figure 6 shows a proposed user interface mock-up of this digital health scorecard visualisation, which has been adapted from Sedhom et al. (2021) [42]. A widget of the main scorecard score results could be embedded into app stores. This would make the data more accessible to users and help inform them in lay terms about the app’s features, utility, and potential risks, empowering end-users to make informed decisions about which apps are most appropriate for their needs. For instance, a clinician will likely place greater value on an app’s clinical and privacy score, whereas a pregnant user may prioritise other domains depending on their needs. 

### 4.4. Study Limitations

This scoping review had several study limitations. Firstly, we were only able to assess commercial apps available on the Apple App Store and Google Play Store, with apps used in randomised clinical trials or those only available via private institutions/hospitals unable to be included. Secondly, apps not available in English were excluded, which limits the applicability of the findings for non-English speaking pregnancy app users. Thirdly, app information and features were assessed by reviewing information provided in app stores, within apps and developer websites; however, there is a possibility that some data were missing. Finally, the ‘user requirements’ domain of the scorecard approach was created by referring to existing literature about the needs of pregnant women and midwives published in 2016 and 2020, respectively. However, given the rapid shift in the healthcare landscape due to the COVID-19 pandemic, especially in relation to digital health, future studies should aim to re-evaluate user needs with reference to more recent literature and/or involve end-users in the design of these user-requirements as carried out by Sedhom et al. (2021) [42]. 

## 5. Conclusions

In this study, we evaluated the quality, credibility, and inclusion of behaviour change techniques of the most popular global commercial pregnancy apps for self-monitoring available in Australia. We used an innovative approach to review these apps by identifying them through web scraping and assessing their quality using the novel scorecard approach. We found that the overall quality, credibility, and behaviour change techniques of the reviewed pregnancy apps to be limited, with significant scope to improve these apps across several domains, most notably in their clinical content and data sharing policies. Finally, we propose a digital health scorecard visualisation that would break down app quality criteria and present them in a more accessible way to clinicians and pregnant users, empowering them to make more informed decisions about which apps are the most appropriate for their needs. 

## Figures and Tables

**Figure 1 ijerph-20-01012-f001:**
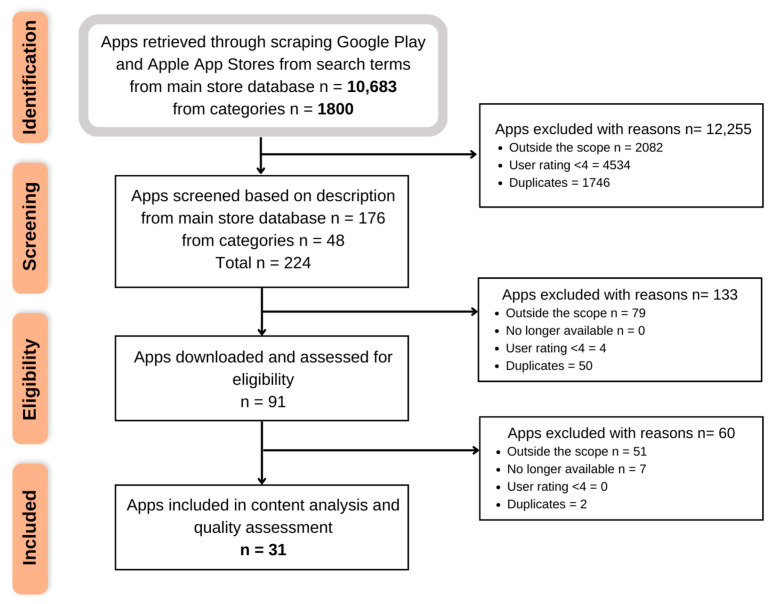
App selection flowchart. Out of the 12,483 pregnancy apps identified, 31 were included for content analysis and quality assessment.

**Figure 2 ijerph-20-01012-f002:**
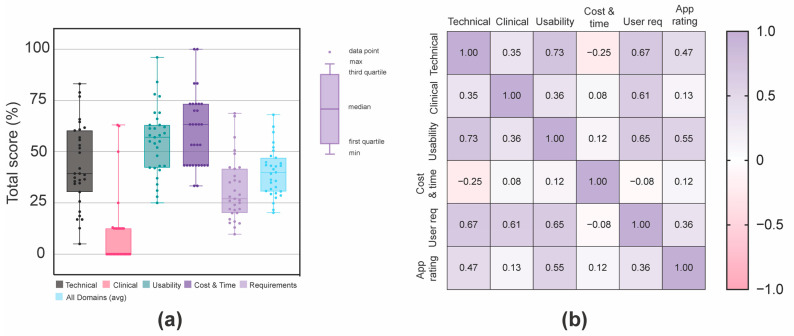
Pregnancy app performance by domain. (**a**) Individual domain scores for each pregnancy app. (**b**) Correlation matrix of each individual domain score and global app ratings. Pearson r values reported. Purple cells denote positive correlations and pink cells denote negative correlations.

**Figure 3 ijerph-20-01012-f003:**
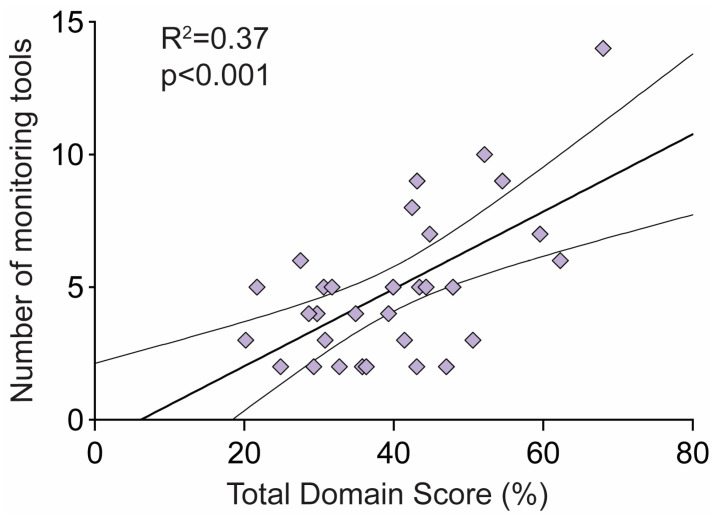
Linear regression between total domain score and the number of monitoring tools for all 31 apps.

**Figure 4 ijerph-20-01012-f004:**
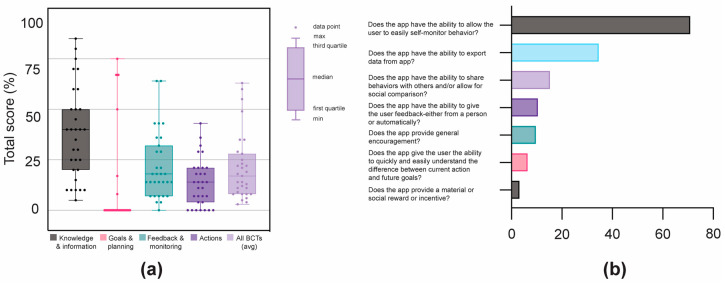
Behaviour change techniques (BCTs) and feedback and monitoring scores. (**a**) Pregnancy app performance by BCT category. (**b**) Breakdown (%) of overall app scores in the ‘feedback and monitoring’ category.

**Figure 5 ijerph-20-01012-f005:**
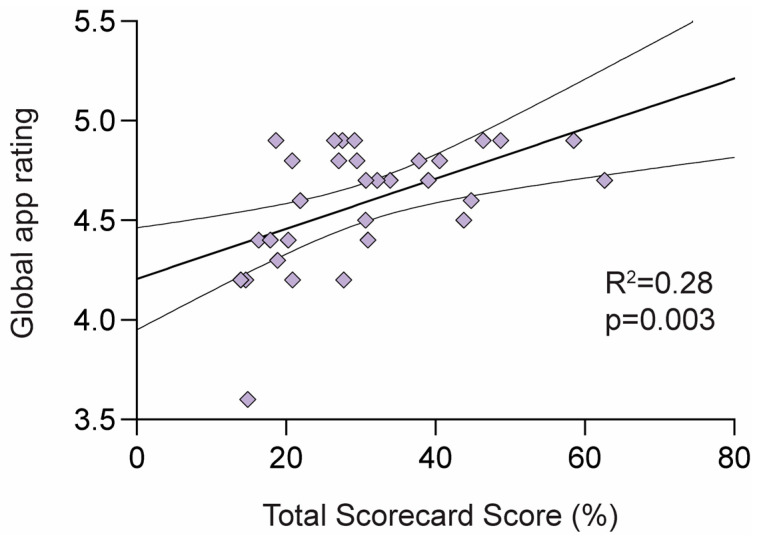
Linear regression between total scorecard score and the global app ratings for all 31 apps.

**Figure 6 ijerph-20-01012-f006:**
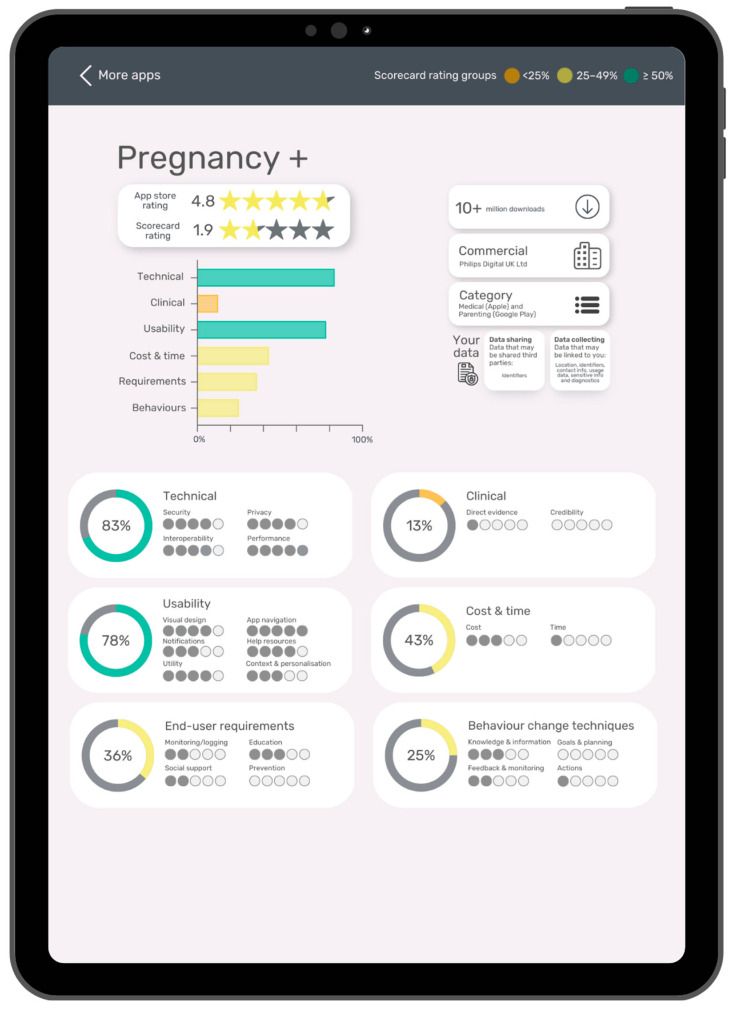
Digital health scorecard: proposed user interface (UI) on a tablet device using the Pregnancy+ app as an example. The global app store rating is listed with the overall scorecard rating (domain and behaviour change technique (BCT) score)). The breakdown of domain and BCT scores is listed below (technical, clinical, usability, cost and time, end-user requirements, and BCTs.

**Table 1 ijerph-20-01012-t001:** App inclusion and exclusion criteria.

Inclusion Criteria	Exclusion Criteria
Cost: free, freemium, and paid Availability: apps available on Apple App Store and/or Google Play Store App popularity: apps with an average user rating of ≥4 and user ratings of at least 20 App focus: apps designed for pregnancy; monitoring of physical health or wellness during pregnancy App features: includes more than one monitoring tool	Cost: no apps were excluded based on their cost. If an app had a free and paid version, the free version was downloaded and tested first. If the paid version allowed for additional self-monitoring app features, the app account was upgraded. App focus: apps designed to focus essentially on fertility, postpartum, or for baby monitoring; apps that did not include features to monitor physical health or wellness or behaviours during pregnancy. App features: app includes no or only one monitoring tool. Apps including just kick or contraction counters were excluded.

**Table 2 ijerph-20-01012-t002:** The 15 pregnancy app monitoring tools found across all apps.

App Name	A	B	C	D	E	F	G	H	I	J	K	L	M	N	O	Total /15
Ovia Pregnancy Tracker																14
Glow/Nurture Tracker																10
Woman’s Pregnancy																9
Pregnancy Tracker-Wachanga																9
Velmio Pregnancy Tracker																8
AMMA Pregnancy Tracker																7
WebMD Pregnancy																7
myFetalLife																6
Baby2Body Pregnancy																6
Pregnancy-Krishnu																5
280 days Pregnancy																5
MomDiary																5
Pregnancy Tracker-Momly																5
Pregnancy-Sevenlogics																5
Sprout Pregnancy																5
Stork Pregnancy Tracker																5
Babynote Pregnancy																4
Hello Belly																4
Pregnancy App-Amila Tech																4
Pregnancy Companion																4
Pregnancy+																4
Happy Pregnancy Ticker																3
Pregnancy Week-Promotube																3
Pregnancy Tracker-Enes																3
BabyCentre																2
My Pregnancy-Aleksei																2
Pregnancy Tracker-Hylal																2
Pregnancy Week-Paydos																2
Pregnancy Tracker-Timskiy																2
Pregnancy Tracker-FitnessLab																2
Pregnancy Tracker-Fittur																2

A = weight, B = symptoms, C = diet, D = diary, E = ultrasound, F = bump, G = counters (kick, contraction), H = exercise (yoga, Kegel), I = mental health (mood, meditation), J = medications, K = sleep, L = blood pressure or heart rate, M = blood glucose, N = doctors’ check-ups, O = temperature; Purple—yes, app does have that monitoring tool and light purple—no, app does not have that monitoring tool. A total of 15 monitoring tools were identified across all apps.

**Table 3 ijerph-20-01012-t003:** The top 10 pregnancy apps. Top 10 pregnancy apps based on their domain (technical, clinical, usability, cost and time, and user requirements) and their behaviour change technique scores compared to their app store rating.

App Name	Total Scorecard Score (avg %)	Scorecard Rating /5	Global App Store Rating /5	Domain Score (avg %)	Behaviour Change Techniques Score (avg %)
Baby2Body	62.6	3.1	4.7	62.3	63.0
Ovia Pregnancy Tracker	58.5	2.9	4.9	68.0	49.0
Velmio Pregnancy Tracker	48.7	2.5	4.9	42.5	55.0
Pregnancy Tracker ~Fittur	46.4	2.3	4.9	32.7	60.0
Woman’s Pregnancy	44.8	2.2	4.6	54.6	35.0
WebMD Pregnancy	43.8	2.2	4.5	59.6	28.0
Glow/Nurture Tracker	40.6	2.0	4.8	52.1	29.0
Pregnancy Tracker—Wachanga	39.1	2.0	4.7	43.1	35.0
Pregnancy+	37.8	1.9	4.8	50.6	25.0
AMMA Pregnancy Tracker	33.9	1.7	4.7	44.8	23.0

## Data Availability

Data supporting reported results are included as Appendix A. All data are available upon reasonable request to corresponding author.

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
