# Peer review of "Pregnancy Apps for Self-Monitoring: Scoping Review of the Most Popular Global Apps Available in Australia"

_ijerph, 2023, doi:10.3390/ijerph20021012_

Round 1

Reviewer 1 Report

Comments to the authors:

The manuscript titled “Pregnancy apps for self-monitoring: scoping review of the most 2 popular global apps available in Australia” aimed to assess the quality of pregnancy apps for self-monitoring, and their usage of Behaviour Change Techniques . Overall it is reasonably well written manuscript and may be accepted with minor changes.

 Following are my comments/questions:

Abstract: A formal format (Aim/goal, study deign, results, conclusion ect) will be more reader friendly.

The authors state that they used the scorecard approach to evaluate the apps, I ma just curious whether any of the apps evaluated also had features to relay information to their health care providers.

Thank you for a comprehensive review. I enjoyed reading it!

Author Response

“The manuscript titled “Pregnancy apps for self-monitoring: scoping review of the most 2 popular global apps available in Australia” aimed to assess the quality of pregnancy apps for self-monitoring, and their usage of Behaviour Change Techniques. Overall it is reasonably well written manuscript and may be accepted with minor changes.

We thank reviewer 1 for the positive evaluation of our manuscript. We have addressed the points raised below. You may also see the attachment.

“Abstract: A formal format (Aim/goal, study deign, results, conclusion ect) will be more reader friendly.”

We agree that the abstract would benefit from formal headings, however the journal author guidelines instruct that “the abstract should be a single paragraph and should follow the style of structured abstracts, but without headings.” Instead, we had added the phrases “the aim of this study”, and “we conclude that…” to make these sections more defined.

“The authors state that they used the scorecard approach to evaluate the apps, I ma just curious whether any of the apps evaluated also had features to relay information to their health care providers.”

Under the technical domain of the scorecard approach, we scored apps on whether they included a data export feature and had support for data integration with other software/systems (e.g., that would allow for data integration with an electronic medical record). We have added these to results section 3.2.2. Technical (lines 317-322):

“Forty-five percent (14/31) of apps provided a means to export user data (4/31 exported data as PDFs and 10/31 exported data as CVSs and/or JSON). None of the apps provided information within the app or app store that they support Fast Healthcare Interoperability Resources (FHIR) Application Programming Interfaces (APIs), which allows for user data to be integrated with other software such as electronic medical records.”

In addition, one of the end-user requirements that we identified for the ‘tracking’ subdomain was whether apps had a feature for users to connect or message their healthcare team. We have added results to section 3.2.5. End-user requirements (lines 358-363):

“Another end-user requirement in the ‘tracking’ subdomain was whether apps had a feature for users to connect or message their healthcare team. Only 26% of apps had a feature to connect users to their healthcare professionals primarily by allowing users to share a ‘health report’ generated by the app via email. Only one app, Ovia Pregnancy Tracker had a feature for users to connect with their healthcare professional/provider if the user has the relevant health insurance plan and lives in the United States.”

Reviewer 2 Report

Dear Authors,

This articles is something new for me.

It is interesting.

In my opinion you should put terms into inverted commas in lines 133-138.

Author Response

This articles is something new for me. It is interesting.

We thank reviewer 2 for the positive evaluation of our manuscript. We have addressed the points raised below. You may also see the attachment.

In my opinion you should put terms into inverted commas in lines 133-138.”

We have now added inverted commas to the search terms used (lines 138-143).

Reviewer 3 Report

The authors presented a review of Pregnancy self-monitoring apps. The paper is well written, and the review methodology is sound. I have only a few comments:

1.     Figure 1: Apps that have a score of less than 4.0 are removed in the first step. Why are another 4 apps, with a score of less than 4.0 removed in the second step as well? Why were these not removed in the first step?

2.     Line 239: Please add that these 15 monitoring tools are shown in table 2.

3.     Usually, papers don’t just write down the values for the “intraclass correlation” but also how the score should be interpreted. (e.g. moderate reliability for values between 0.5 and 0.75 indicate).

4.      I cannot view the supplementary materials. Considering that Table S1 is relatively crucial and part of the main contribution of the paper, it might be worth adding the relevant table to the appendix instead.

Author Response

“The authors presented a review of Pregnancy self-monitoring apps. The paper is well written, and the review methodology is sound. I have only a few comments.

We thank reviewer 1 for the positive evaluation of our manuscript. We have addressed the points raised below. You may also see the attachment.

“1. Figure 1: Apps that have a score of less than 4.0 are removed in the first step. Why are another 4 apps, with a score of less than 4.0 removed in the second step as well? Why were these not removed in the first step?”

When the second round of screening was conducted, we noticed that 4 of the apps were no longer rated 4 and above in the app store and were excluded. The following section has been added to the results to clarify this (lines 231-236):

“After removing duplicates, apps with a user rating <4 and apps outside the scope, the remaining commercial apps (n=224) were then screened based on their description and downloaded and assessed for eligibility (n=91). Four of these apps were found to no longer have a user rating of 4 or above and were excluded. Finally, thirty-one apps were deemed eligible for inclusion in this review and were assessed for content analysis and quality and behaviour change techniques (Figure 1).” 

“2. Line 239: Please add that these 15 monitoring tools are shown in table 2.”

We have modified the text to more clearly indicate that the 15 monitoring tools are shown in table 2. (lines 248-252):

“Across the 31 apps reviewed, 15 different monitoring tools were identified. The average number of monitoring tools included in each app was 5 (range 2-14). The breakdown of the monitoring tools contained within each app is shown in Table 2.”

“3. Usually, papers don’t just write down the values for the “intraclass correlation” but also how the score should be interpreted. (e.g. moderate reliability for values between 0.5 and 0.75 indicate).”

A statement has been added to explain how the intraclass correlation should be interpreted (lines 202-205):

“Inter-scorer reliability was evaluated by running an intraclass correlation across multiple Scorecard and BCT domains and was found to be 0.72 F(359,355(=6.2, p<0.001). An intraclass correlation less than 0.5 indicates poor reliability, between 0.5 and 0.75 indicates moderate reliability, between 0.75-0.9 indicates good reliability, and above 0.9 indicates excellent reliability.”

“4. I cannot view the supplementary materials. Considering that Table S1 is relatively crucial and part of the main contribution of the paper, it might be worth adding the relevant table to the appendix instead.”

We are sorry that the reviewer was unable to view the supplementary material. We will make sure that it is readily accessible via the supplementary material.